# Community engagement in research addressing infectious diseases of poverty in sub-Saharan Africa: A qualitative systematic review

Zewdie Birhanu Koricha[1,2☯], Yosef Gebreyohannes Abraha[1,3,4☯]*, Sabit Ababor Ababulgu[1,3,4☯], Gelila Abraham[1,2,5☯], Sudhakar Morankar[1,2☯]

1 Public Health Faculty, Department of Health, Behaviour and Society, Jimma University, Jimma, Ethiopia, 2 Ethiopian Evidence-Based Healthcare and Development Centre: a JBI Centre of Excellence, Jimma University, Jimma, Ethiopia, 3 Ethiopian Public Health Institute, Knowledge Translation Directorate, Addis Ababa, Ethiopia, 4 The Ethiopian Public Health Institute, Ethiopian Knowledge Translation Group for Health: a JBI Affiliated Group, Addis Ababa, Ethiopia, 5 Health Policy & Management Department, Jimma Institute of Health, Jimma University, Jimma, Ethiopia

☯ These authors contributed equally to this work.
* yosephgy@gmail.com

## Abstract

Though engaging communities in research processes has several advantages and implications, research efforts are poorly embedded in and linked with communities, especially in low- and middle-income countries (LMICs). There is also a need for more empirical evidence on effectively engaging communities in research in LMICs, specifically in Sub-Saharan Africa (SSA). Thus, there is an urgent need to synthesize existing evidence on community engagement experiences in research in SSA. Therefore, this review aimed to synthesize the existing community engagement experiences and related barriers to engaging communities in health research focusing on infectious diseases of poverty in SSA. The systematic review was conducted following the JBI methodology for qualitative systematic reviews. The review included both published and unpublished studies. A thematic analysis approach was used for data synthesis. A total of 40 studies were included in the review. Community engagement in the conceptualization of the research project, analysis, dissemination, and interpretation of the result was rare. On top of this, almost all the research projects engaged the community at a lower level of engagement (i.e., informing or consulting the community at some point in the research process), suggesting the importance of integrating communities in the entire research cycle. The lack of shared control over the research by the community was one of the significant challenges mentioned. This review uncovered that community engagement in the research process is minimal. Nevertheless, the review generated valuable evidence that can inform researchers and research stakeholders to promote effective community engagement in the research process addressing infectious diseases of poverty. Despite these, it requires rigorous primary studies to examine the applicability and usefulness of community engagement, including developing valid metrics of engagement, standardizations of reporting community engagements, and views

**Data Availability Statement:** This is a systematic review, and all the results are reported in the manuscript.

**Funding:** This work was supported by a grant from the World Health Organization, a Special Program for Research and Training in Tropical Diseases (2021/1135757-0 to Jimma University). The funders had no role in study design, data collection and analysis, decision to publish, or preparation of the manuscript.

**Competing interests:** The authors have declared that no competing interests exist.

and understandings of communities and stakeholders on the values, expectations, and concepts of community engagement in research.

## Introduction

Diseases of poverty (also known as poverty-related diseases) are diseases that are more prevalent in low-income populations [1]. They include the three primary poverty-related diseases (i.e., tuberculosis, HID/AIDS, and malaria), neglected tropical diseases (NTDs), as well as other diseases related to poor health behaviour. These diseases are markers of extreme poverty and inequality propagated by political, economic, social, and cultural systems that affect health and well-being. Thus, these diseases of poverty remain endemic to many countries of sub-Saharan Africa (SSA) that are left behind in socioeconomic progress [1,2].

Community-engaged research is critical in overcoming poverty-related infectious diseases [3,4]. Community engagement (CE) in research is defined as the process of working collaboratively with and through groups of people affiliated by geographic proximity, special interest, or similar situations to address issues affecting the well-being of those people [5]. It encourages the community to participate in addressing its own health needs and ensures that researchers understand community priorities [6,7]. Community-engaged research projects create a dynamic and interactive relationship between researchers, policy-makers, and the community for co-creation and translations of knowledge for better health outcomes [3,8–10]. It advocates research participants as partners rather than merely trial subjects or eventual users of the interventions. Several studies also indicated that communities engaged in various clinical trials, from protocol development to dissemination of results, to ensure the research was scientifically appropriate and ethically sound [11–20].

Although the focus in health research may be shifting from infectious to non-communicable diseases, the infectious diseases of poverty remain a major burden of global health concern, especially in low- and middle-income countries (LMICs) [2]. There is also a long way to go to advance research and evidence-based public health interventions in LMICs, especially in SSA [1,2]. On top of limited research outputs in these settings, the research practices often do not effectively engage the target community, limiting the policy influence of research evidence to improve population health outcomes [3].

However, despite the potential benefits of community engagement in research, communities lack adequate and sustained engagement in research addressing infectious diseases of poverty, especially in SSA. This leads to poor alignment of research with community needs and priorities, low uptake of research evidence for decision-making, and limited health improvements and capacity building of individuals and institutions [21–30]. Furthermore, there is a lack of empirical evidence on how to engage communities in research in these settings and what barriers and facilitators hinder community engagement in research [31].

Therefore, this review aims to summarize existing evidence on experiences and practices in community engagement in infectious diseases of poverty research in sub-Saharan Africa and to identify challenges and lessons learned from engaging communities in health research. To achieve this goal, this paper addresses the following research questions:

1. What are community engagement activities and experiences in research for addressing infectious diseases of poverty in SSA?

2. What are the challenges encountered and the mitigation strategies in community-engaged research addressing infectious diseases of poverty in SSA?

## Methods

The systematic review followed the Joanna Briggs Institute (JBI) methodology for qualitative systematic reviews [32]. JBI is an international collaboration of health scientists, professionals, and researchers committed to 'Best Practice' at the University of Adelaide. The authors selected this methodology because it provides a comprehensive guide to conducting different types of systematic reviews, including qualitative systematic reviews. In addition, the JBI software (JBI SUMARI), designed to guide the reviewer in the systematic review, from protocol development to the final report stage, and to archive all review components, provides end-to-end support in conducting this review. This review was performed based on an a priori protocol available at (https://osf.io/tdjbp/?view_only=7b930ca19f0b48deacb697e55bf63bf0).

### Inclusion criteria

The eligibility criteria were based on the JBI framework for undertaking a qualitative systematic review (i.e., Participant, Phenomena of Interest, and Context; PICo) [32]. The PICo mnemonic aligns with the review questions for this review. It provides potential readers with significant information about a review's focus, scope, and applicability to their needs.

### Participants (P)

For this paper, we define community in terms of geography and relationships and refer to a group of people who reside in a specific location and the relationships between them [33]. Thus, this review considered studies that include any individual or group participating in any stage of the research process. Studies that report community participation only as a trial subject or simply seeking informed consent to inform participants about the study were excluded from the review.

### Phenomena of interest (I)

The review considered studies investigating the experiences, practices, and strategies or approaches for engaging communities in research. The review also considered studies that identify the core challenges or barriers and lessons of effective community engagement in research to address infectious diseases of poverty. We considered studies that reported community engagement activities or experiences in at least one of the following research processes [6,34,35]: a) defining or identifying the research problem; b) choosing research methods; c) developing sampling procedures; d) designing interviews and survey questions or any research tool or formats; e) recruiting study participants; f) collecting data; g) analysing collected data; h) interpreting study findings; i) writing reports (any form such as a technical report, summary/abstract, etc.); j) giving or attending presentations at meetings and conferences; k) implementing the intervention; and l) translating research findings into policy products such as policy briefs, guidelines, etc.

### Context (Co)

The review focused on synthesizing available evidence from countries in the Sub-Saharan African (SSA) region [36].

### Search strategy

Quantitative and qualitative data were considered to address the review's objectives. A three-step search strategy was utilized in this review. First, a limited search of MEDLINE (PubMed) and EMBASE was undertaken to identify published articles. The text words contained in the

titles and abstracts of relevant articles and the index terms used to describe the articles were used to develop a complete search strategy for PubMed in the preliminary stage (see S1 Table). The reference list of all included sources of evidence was screened for additional studies. The review was restricted to studies and reports published in the English language. This is because no review team member can access and retrieve other languages. We limited our search to include studies published or conducted since January 2005 to capture the current practices and experiences in engaging communities in research to address infectious diseases of poverty in SSA. The databases searched include Cochrane Central Register of Controlled Trials (CEN-TRAL), MEDLINE (via PubMed), EMBASE (via Ovid), CINAHL (EBSCO), and Web of Science. To enhance the comprehensiveness of the review, we have also included non-peer-reviewed sources (unpublished studies and grey literature) from Google Scholar and websites of different national and international organizations/institutions.

## Study selection process

Following the search, all identified citations were collected and uploaded into EndNote X9 2018 (Clarivate Analytics, PS, USA), and duplicates were removed. Titles and abstracts were then screened by two independent reviewers (YG & SA) for assessment against the inclusion criteria for the review. Potentially relevant studies were retrieved in full-text, and their citation details were imported into the JBI System for the Unified Management, Assessment and Review of Information (JBI SUMARI) (JBI, Adelaide, Australia) [37]. Two independent reviewers assessed the full text of selected citations in detail against the inclusion criteria. The systematic review recorded and reported reasons for excluding full-text papers that did not meet the inclusion criteria. Any disagreements between the reviewers at each stage of the selection process were resolved through discussion or with additional reviewers (ZB, MS, & GA). The search results and the study inclusion process were reported in the final systematic review and presented in a Preferred Reporting Items for Systematic Reviews and Meta-analyses (PRISMA) guideline [38]. See the S1 Checklist, PRISMA, for more details.

## Assessment of methodological quality

Two independent reviewers assessed the studies for methodological validity before inclusion in the review using a standardized critical appraisal instrument from the Joanna Briggs Institute Qualitative Assessment and Review Instrument (JBI-QARI) (see S2 Checklist). Any disagreements with the reviewers (YA and SA) were resolved through discussion or with a third reviewer (ZB). This occurred 6 times during the methodological quality assessment, and three disagreements were resolved with a third reviewer (ZB). All included studies were assessed using JBI-QARI, and their overall critical appraisal scores are presented. Regardless of their methodological quality results, studies were included, given that they were relevant to the area of inquiry.

## Data extraction

Two reviewers (YA & SA) independently extracted data from the included studies. Qualitative data were extracted from studies included in the review using the standardized data extraction tool from JBI-QARI (see S3 Checklist). The data extracted included specific details about the phenomena of interest, populations, study methods, key findings, and outcomes of significance to the review question and specific objectives. The key findings were broadly related to any findings related to community engagement, such as engagement methods, approaches, strategies, lessons, barriers, and facilitating factors. Any reviewer disagreements were resolved through discussion or with a third reviewer.

### Data synthesis

The review team planned a mixed-method systematic review (MMSR) to produce a more comprehensive evidence synthesis by combining and integrating qualitative and quantitative studies [32]. However, due to the limited availability of quantitative studies relevant to the objective of our review, we adapted our approach to qualitative synthesis methods. Accordingly, studies included in this review used a range of qualitative findings that address a variety of experiences. Nevertheless, pooling the results using the statistical meta-analysis and meta-aggregation was impossible. We used a thematic synthesis approach described by Thomas and Harden (2008) [39]. A thematic synthesis helps analyze data from qualitative evidence syntheses exploring people's perspectives and experiences, acceptability, appropriateness, and factors influencing implementation. Thematic synthesis offers a flexible, systematic, and transparent method to move from the findings of multiple qualitative studies to synthesis. Finally, the findings were synthesized manually, and the results are presented in narrative form.

We used Hanacek's (2010) guide [34] to help identify and categorize the different phases of conducting health research. These emergent findings were classified into five stages of research processes: 1) research conceptualization, 2) research design, 3) research implementation, 4) data analysis and interpretation, and 5) dissemination and translation. Consequently, for the stage of the research process, a score of "1" or zero (0) was assigned for each included study. This means that when community engagement exists at each stage, a score of 1 is assigned, making a maximum score of five (5) and a minimum score of zero (0). The higher score implies that the community was engaged in most of the research processes and vice versa.

Further, we used the WHO community engagement framework [40] to classify the level of community engagement in each included study. This engagement framework consists of five levels of community engagement in any research stage, ranging from informing (reaching out) to shared leadership or empowering the community to make decisions. Thus, community engagement activities in each study process were assigned to one of the levels of community based on the detailed descriptions outlined in Table 1 below [5,6,40].

## Results

### 1. Description of studies

There were 2,259 studies identified as potentially relevant to the review (2,230 from databases, 3 from reference lists, and 26 from organizational websites). The last search was done on July 10, 2023. After removing duplicates, 1955 records were included for initial screening. Of these, 1885 records were excluded after screening by title and abstract based on the eligibility criteria, leaving 70 records considered in more detail. The full text was retrieved for 69 records. All these records were reviewed against the eligibility criteria set for the review. A further 29 records were subsequently excluded for not meeting the relevant criteria, and the reasons for exclusion are reported in the supplementary file (see the S1 Text). Of the 29 excluded records, two studies [41,42] were excluded after two reviewers appraised the studies for methodological quality. After a quality appraisal, 40 studies that met the inclusion criteria were included in the final review. The search and the study selection process are illustrated in Fig 1.

### 2. Characteristics of included studies

This section provides an overview of the included studies. Refer to the (S2 Table) for a detailed study description.

**Study designs.**   Except for two, all these included studies were qualitative. Two of the included studies [17,19] were mixed-method studies and included in the review's qualitative

**Table 1. Elaborated operational definitions for levels of community engagement in research.**

| Level of engagement | Descriptions of community engagement in research fulfilling the criteria |
|---|---|
| Informing | This happens when there is minimal community engagement in the research process (e.g., being informed about the research to be conducted); communities are provided only with information (communication flows from the one who conducts the research to the community, just for the sake of informing). At this level, the entities coexist, but optimally, it establishes communication channels and channels for outreach. |
| Consulting | This occurs when there is more community engagement in any research process (e.g., getting feedback on study procedures and data collection instruments from the community). Communication flows to the community and then back to the researcher to seek answers. Communities are allowed to give feedback. At this level, the entities share information and develop connections. |
| Involving | This happens when there is better community engagement in the research process (e.g., communities participate in designing data collection tools, collecting data via surveys or qualitative interviews and data analysis, communities' intervention delivery, monitoring, and evaluations etc.). Communication flows both ways in a participatory form. It involves more community participation on issues related to the research process. At this level, the entities cooperate, and partnerships are established with increased cooperation. |
| Collaborating | This occurs when community engagement is in the research process (e.g., community-based participatory approaches to designing user-friendly forms or strategies or interventions, co-creating and contextually adapting tools, etc.). Communication flow is bidirectional. The entity forms partnerships with the community on each aspect of the research project, from development to solution. The entities form bidirectional communication channels at this level and build trust and partnerships. |
| Empowering | Empowerment is considered a level with the highest degree of participation, where the target communities and beneficiaries collectively gain greater control over decisions and actions in the entire research process. Inherently, this involves changes in power dynamics whereby communities or individuals take control of the final decision-making in the research process. At this level, entities form a strong partnership structure affecting the broader community and with bidirectional solid trust. |

component. The studies consisted of a variety of methodologies. Thirty-two studies were descriptive qualitative [11,13,14,16–19,43–67] and adopted a qualitative data collection and analysis approach. The remaining studies used other forms of qualitative approach: four were based on grounded theory [68–71], two were content analysis [15,72], one was formative [12], and one was narrative [73]. A variety of data collection methods was also used. Eight included studies followed a community-based participatory action research [44,46,48–50,58,60,66].

**Focus area by disease (type of infectious disease of poverty).**   Nineteen studies [11,12,14,15,48,49,53–56,59,60,62–65,67,72,73] provided data about community engagement in HIV/AIDS-related research, while 14 studies [9,12–15,46,47,50,54,60,69,72–74] provided data on malaria-related research. The remaining studies provided data on Schistosomiasis [46,58], Ebola [45,68], tuberculosis [50], Lymphatic Filariasis [61], and Typhoid [52].

**Participants.**   These studies included a wide variety of participants, including representatives of the community, community advisory board members, community leaders, religious leaders, traditional practitioners, community health committee members, parents, students, school authorities, researchers, etc.

**Country of research/context.**   The review focused on community engagement activities in research to address the infectious diseases of poverty in SSA countries. The majority of the studies were conducted in Kenya (n = 14 studies) [11,14,16,17,49,51,53,55,57,59,63–65,72] and South Africa (n = 10 studies) [11,15,48,50,60,62,64,67,72,73]. Other countries include Tanzania (n = 7 studies) [11,13,16,46,64,67,70], Uganda (n = 5 studies) [16,56,64,66,67], Malawi (n = 4 studies)[11,12,16,52], Nigeria (n = 4 studies) [44,58,64,69], Mozambique (n = 3 studies) [11,44,67], Zambia (n = 3 studies) [61,62,67], Mali (n = 3 studies) [13,43,47], Liberia (n = 3 studies) [45,68,71], Burkina Faso (n = 2 studies) [13,18], Zimbabwe (n = 2 studies) [11,54],

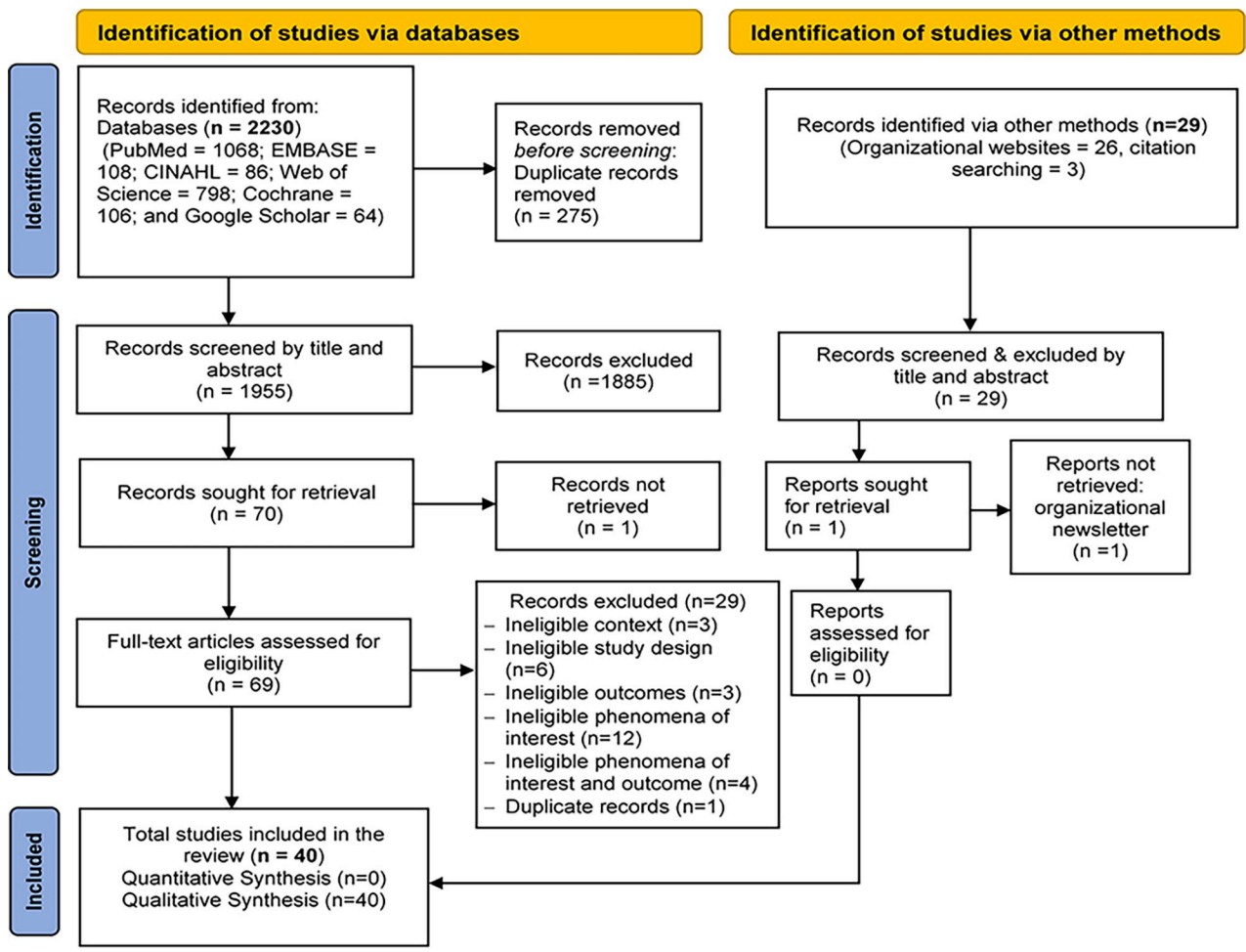

**Fig 1. PRISMA flowchart of the study selection process.**

Democratic Republic of Congo (n = 2 studies) [44,64] and Gambia (n = 1 study) [19]. Nine of the included studies were multinational [11,13,16,44,45,62,64,67,72].

**Phenomena of interest.**   Concerning specific phenomena of interest addressed by the studies, all the included studies reported community engagement experiences for addressing infectious diseases of poverty in any research process and documented lessons learned and facilitators for community engagement in research [11–19,43–73]. Of the included articles, 18 studies [11,13,14,16–19,49,51–53,56–58,60,62,64,73] stated the challenges or barriers to community engagement in research.

## 3. Methodological quality of included studies

The methodological quality of the qualitative studies was performed using the JBI-QARI quality appraisal tool. A total of 40 qualitative studies were assessed for methodological quality and included in the final synthesis. Most of the studies were deemed methodologically rigorous enough concerning the assessment criteria (scored between two and eight against the 10 critical appraisal questions applicable to qualitative studies). The reviewers did not assign a "not applicable" rating to any qualitative study assessment criteria. Studies were not excluded because of their methodological quality. Hence, all studies were included for further synthesis, irrespective of their quality. The critical appraisal scores are outlined below in Table 2.

**Table 2. Critical appraisal results for included studies.**

| Reference | Q1 | Q2 | Q3 | Q4 | Q5 | Q6 | Q7 | Q8 | Q9 | Q10 | Score |
|---|---|---|---|---|---|---|---|---|---|---|---|
| Agot et al. 2019[72] | Y | Y | Y | Y | Y | N | U | Y | Y | Y | 8/10 |
| Angwenyi et al. 2014[17] | Y | Y | Y | Y | Y | N | U | Y | Y | Y | 8/10 |
| Bandewar, Kimani and Lavery 2010[65] | Y | Y | Y | U | Y | N | U | U | Y | Y | 6/10 |
| Beard et al. 2020[73] | Y | Y | Y | Y | Y | N | U | Y | U | Y | 7/10 |
| Broder et al. 2020[11] | Y | Y | Y | Y | Y | N | U | Y | U | Y | 7/10 |
| Corneli et al. 2007[12] | Y | Y | Y | Y | Y | N | Y | U | Y | U | 7/10 |
| Davies et al. 2012[66] | U | Y | Y | Y | Y | N | U | U | Y | Y | 6/10 |
| Denison et al. 2017[67] | Y | Y | Y | Y | Y | U | U | Y | U | Y | 7/10 |
| Diallo et al. 2005[43] | Y | Y | Y | Y | Y | N | Y | U | U | U | 6/10 |
| Dierickx et al. 2018[19] | Y | Y | Y | Y | Y | N | U | Y | Y | Y | 8/10 |
| Doshi et al. 2017[14] | Y | Y | Y | Y | Y | N | N | Y | Y | Y | 8/10 |
| Faye and Lugand 2021[44] | Y | Y | Y | Y | Y | N | U | Y | Y | Y | 8/10 |
| Folayan et al. 2019[45] | Y | Y | Y | U | Y | N | U | U | Y | Y | 6/10 |
| Freudenthal et al. 2006[46] | Y | Y | Y | Y | Y | N | U | U | Y | Y | 7/10 |
| Hartley et al. 2021[47] | Y | Y | Y | U | U | N | U | U | Y | Y | 5/10 |
| Hullur et al. 2016[48] | Y | Y | Y | U | U | N | U | U | Y | U | 4/10 |
| Kamanda et al. 2013[49] | Y | Y | Y | Y | Y | N | Y | U | Y | Y | 8/10 |
| Mabunda et al. 2016[50] | Y | Y | Y | Y | Y | N | U | Y | Y | Y | 8/10 |
| Marsh et al. 2011[51] | Y | Y | Y | Y | Y | N | Y | U | U | Y | 7/10 |
| Martínez et al. 2018[68] | Y | Y | Y | Y | Y | N | U | Y | Y | Y | 8/10 |
| Meiring et al. 2019[52] | U | Y | Y | U | U | N | U | Y | U | Y | 4/10 |
| Molyneux et al. 2016[53] | U | Y | Y | U | U | N | U | U | N | U | 2/10 |
| Morin et al. 2008[54] | Y | Y | U | Y | Y | N | U | U | Y | Y | 6/10 |
| Mtove et al. 2018[16] | Y | Y | Y | Y | Y | N | U | Y | Y | Y | 8/10 |
| Nakalega et al. 2021[55] | Y | Y | Y | U | U | N | U | U | Y | Y | 5/10 |
| Nakibinge et al. 2009[56] | Y | Y | Y | Y | Y | N | Y | U | U | Y | 7/10 |
| Nyika et al. 2010[13] | Y | Y | Y | Y | Y | N | U | U | Y | Y | 7/10 |
| Ogunrin et al. 2021[69] | Y | Y | Y | Y | Y | N | U | Y | Y | Y | 8/10 |
| Okello et al. 2013[57] | Y | Y | Y | Y | Y | N | U | Y | Y | Y | 8/10 |
| Olaseha and Sridhar 2005[58] | Y | Y | Y | Y | Y | N | Y | U | U | Y | 7/10 |
| Pare et al. 2021[18] | Y | Y | Y | Y | Y | N | U | Y | Y | Y | 8/10 |
| Reddy et al. 2010[15] | Y | Y | Y | Y | Y | N | U | Y | Y | Y | 8/10 |
| Rennie et al. 2017[59] | Y | Y | Y | Y | Y | N | U | Y | U | Y | 7/10 |
| Reynolds et al. 2011[70] | Y | Y | Y | Y | Y | N | U | U | U | U | 5/10 |
| Shahmanesh et al. 2021[60] | Y | Y | Y | U | U | N | U | U | Y | Y | 5/10 |
| Silumbwe, Halwindi and Zulu 2019[61] | Y | Y | Y | Y | Y | N | U | U | Y | Y | 7/10 |
| Simwinga et al. 2016[62] | Y | Y | Y | Y | Y | N | Y | U | Y | Y | 8/10 |
| Tarr-Attia et al. 2018[71] | Y | Y | Y | Y | Y | N | U | Y | Y | Y | 8/10 |
| Vreeman et al. 2012[63] | Y | Y | U | Y | Y | N | U | U | Y | Y | 6/10 |
| Yotebieng et al. 2019[64] | Y | Y | Y | Y | Y | N | U | Y | U | Y | 7/10 |

Y, yes; N, no; U, unclear.

## 4. Review findings

Based on the review objective, the findings are organized under three major themes and domains using the following structure:

1. Community engagement activities or experiences in health research addressing infectious diseases of poverty

2. Barriers faced and their mitigation strategies during community engagement in health research

Within each central theme, subthemes were created to present the review findings meaningfully. The following sections explored each review finding under these themes and domains.

**4.1. Community engagement activities in research.**   A total of 35 studies [11–19,43–52,54–56,58–67,71–73] reported community engagement activities or experiences for addressing infectious diseases of poverty in any research process (see S3 Table for the details). These community engagement activities are organized into five overarching themes following Hanacek's (2010) [34] phases of the research process. These phases for categorizing community engagement in the research process include research conceptualization, designing, conducting/implementing the research, data analysis and interpretation, and disseminating and translating research findings. These themes were used to document the community engagement activities for addressing infectious diseases of poverty and are described below.

*4.1.1. Community engagement in research conceptualization.* Only eleven studies of the included studies (11/40) reported evidence of community engagement in the formulation of the research problem or research questions; protocol development; searches and reviews of the literature relating to the research problem and developing a framework; and research agenda-setting or priority setting [11,12,43,45,46,48,49,58,64,66,67].

Of the included studies, only five [11,12,43,46,49] provide information on the engagement of the community members related to the study protocol. These studies indicated that community members have provided guidance and input to the study protocol development that helps to modify the research protocol to support cultural acceptability while maintaining study objectives. For instance, Broder et al. (2020) [11] explored community engagement in Antibody Mediated Prevention (AMP) studies, also known as HIV Vaccine Trials Network (HVTN) and the HIV Prevention Trials Network (HPTN) studies. In these studies, community members were engaged through their community advisory structures, such as community working groups and site Community Advisory Boards (CABs), which ensured the principles of community involvement, facilitated community participation throughout the entire research process, and served as the voice for the community and study participants at different levels. At the protocol-specific level, community working groups (composed of one community educator and one community advisory board representative from each participating site) provided guidance and input to the study protocol development team.

At a site-specific level, CABs brought specific and unique expertise to the research process by defining the scientific agenda, representing the community, and helping in the enrolment of participants by raising research-related issues or concerns that may impact the local participants, local community, or the study overall. Notably, members of the AMP community working groups participated in regular tele/video conference calls, face-to-face meetings, protocol development team meetings, site assessment visits, study-specific consultations, training, and workshops. They freely expressed any concerns about the study's conduct without fear of repercussions [11].

Formative research commenced by Corneli et al. (2007) [12] to inform the design of a clinical trial in the breastfeeding, antiretroviral, and nutrition (BAN) in Malawi involved community members to ensure participants' understanding of the research, safeguard participants, and increase the feasibility and acceptance of the clinical research in the community. The

engagement enabled the research team to rapidly modify the protocol and achieve cultural acceptability while maintaining study objectives.

A study by Freudenthal et al. (2006) [46] describes community engagement in school-based participatory action research to prevent schistosomiasis in northern Tanzania. Researchers established good links with the community members and sensitized the participatory action research concepts and methods, emphasizing dialogue rather than imparting and imposing expert knowledge. At the initial stage, researchers and community members arranged different meetings and workshops in schools and the community. The problem was then identified, i.e., high prevalence of Schistosoma among schoolchildren. They proposed investigating sustainable ways to prevent the disease [46].

Another participatory research identified by Davies et al. (2012) [66] presented the involvement of community members in the design of adverse event forms for real-world reporting to capture information on events associated with artemisinin combination therapies (ACTs) for malaria treatment. In the study, target audiences (community members representing the community) were engaged in defining the problem. During the conceptualization phase, informal discussions between participants and stakeholders were held; key issues with the current form were identified; and the concept was developed with the research team (using a storyboard, diary table, and key fields for inclusion) [66].

In six studies, community members participated in identifying and defining the scientific agenda or health problem [11,48,49,58,66,67]. Denison et al. (2017) [67] identified youth engagement in developing an implementation science research agenda on adolescent HIV testing and care linkages in sub-Saharan Africa. Youth living with HIV (YLHIV) participated in a 2-day meeting with experts in implementation science research agenda setting on a project called Supporting Operational AIDS Research (Project SOAR). YLHIV shared their views and experiences with the meeting participants, voted on the priority research questions, and influenced working group discussions [67].

Research that adopted a community-based participatory research approach by Hullur et al. (2016) [48] explored the community perspectives on HIV, violence, and health surveillance in rural South Africa. In this study, the community participated in identifying and defining health problems. Kamanda et al. (2013) [49] described how community-based participatory research (CBPR) approaches and principles can be incorporated and adapted into the study design and methods of a longitudinal epidemiological study in sub-Saharan Africa. The orphaned and separated children in Kenya were engaged in the community throughout the research project. The community members identified key questions and priorities. Community members gathered *mabaraza* (the traditional form of community assembly in East Africa) and participated in community meetings to discuss the project's feasibility [49].

Participatory action research by Olaseha and Sridhar (2005) [58] explored community engagement in controlling urinary Schistosomiasis. The community members in Ibadan, Nigeria, felt that Schistosomiasis was their primary health problem and listed it as their priority. They systematically pursued the relevant authorities to control the disease [58]. Both Folayan et al. (2019) [45] and Yotebieng et al. (2019) [64] explored community engagement by participating in a consensus-reaching process using Delphi techniques to set research priorities.

*4.1.2. Community engagement in developing research design and strategy.* A substantial number of included studies (21/40) [11,13–17,43,46,49,50,55,58–63,65,66,71,73] reported community engagement in preparing a general research plan, including choosing research methods, selecting or recruiting study participants, developing sampling procedures, designing interviews and/or survey questions or any research tool or format, including consent.

For instance, community members have identified and recruited study participants [11,13–15,17,43,46,49,50,55,61–63,65,73]. In one study, a community advisory board (CAB) assisted in the recruitment of adolescent girls and young women (AGYW) in a study involving adolescents in biomedical HIV prevention research [55].

In another study by Broder et al. (2020) [11], the community working groups participated in adapting consent forms for local use and development of other study-related materials, enhanced the capacity of community members through engaging in protocol-specific training and regional workshops and informing strategies for recruitment and retention. In this study [11], community educators were responsible for developing and implementing site-specific community engagement work plans that outline goals, objectives, and the local scope of work based on a local community needs assessment. Community educators were employed locally by the clinical research sites and were typically full-time employees. They collaborate with their local CAB representatives to assess and identify appropriate educational strategies/materials that need to be developed to educate their communities about the research agenda.

Besides, community representatives contributed to developing local recruitment plans and identifying recruitment strategies (e.g., being present in community locales for face-to-face outreach and advertising on internet dating websites). The recruitment strategies covered a wide range of strategies needed to effectively reach marginalized populations at increased risk for HIV acquisition and address regional differences and the unique cultural nuances of these diverse populations. The strategies were all quite efficient and performed above expectations [11].

A study by Nakalega et al. (2021) [55] described community perspectives on ethical considerations that involve adolescent girls in biomedical HIV prevention research. Two stakeholder meetings were held with adolescent girls and young women (AGYW) aged 16 to 21, purposively selected by the adult and youth community advisory board (CAB) to represent diverse views and age groups.

Nyika et al. (2010) [13] described the case study engaging diverse communities participating in clinical trials. The study indicated communities were involved in various sites, including ethical and regulatory approval of the intended clinical trial in each country; administrative approval from local government structures such as local district health offices; permission to enter the community (community leaders approached to initiate engagement with the community); and recruitment of individual participants.

A study by Beard et al. (2020) [73] documented the experiences of community engagement in the Amajuba Child Health and Wellbeing Research Project, which measures the impact of orphaning due to HIV/AIDS on South African households. At the initial phase of the project, a field office opened, and community members were hired as research assistants and office administrators; a community advisory committee was established to oversee the research activities, and principals and teachers from local schools assisted with the identification of orphans and children at risk of being orphaned and their caretakers invited to participate in the study. In addition, the study engages the community in an ongoing discussion about the study's purpose and procedures, such as giving input elicited on survey instruments before initial piloting and between rounds of data collection [73].

Kamanda et al. (2013) [49] explored the community engagement activities in the OSCAR project. In the project, the research team developed a community advisory board (CAB), and the CAB reviewed mabaraza feedback and approved the study procedures and data collection instruments considering the local procedures and cultural context. Besides, community members were integrated into the research team and identified households caring for orphans and separated children. Community members participated in the monthly meetings at the children's services forum, annual mabaraza, and quarterly CAB meetings [49].

Community representatives contributed to developing local recruitment plans and identifying recruitment and retention strategies [11]. Doshi et al. (2017) [14] used a social network-based approach to contextualize willingness to participate in future HIV prevention trials. The study employed snowball sampling to recruit men who have sex with men (MSM) into the study from Kisumu, Mombasa, and Nairobi. The Kenyan MSM research team comprised seven community researchers selected from the respective MSM communities. Frontline organizations working with MSM were engaged to help identify community researchers. Those chosen as community researchers were well-respected leaders in their communities, experienced in sexual health research and/or programming, and played a central role in the design of data collection tools. As members of the sex worker community, the community researchers were familiar with and sensitive to the lived experiences of the participants [14]. Community researchers who were respected leaders in their communities and experienced in sexual health research and programming played a central role in the design of data collection tools in the study that investigates the willingness of men who have sex with men (MSM) to participate in HIV vaccine efficacy trials [14].

A study by Freudenthal et al. (2006) [46] described that pupils were involved in a household survey as researchers and were change agents in participatory action research in primary schools aimed to create enabling environments for schoolchildren and other community members to adopt practices relevant to reducing the transmission of schistosomiasis. Reddy et al. (2010) [15] described community advisory boards' operations in HIV/AIDS vaccine trials. CABs participated in recruiting youth by consulting staff of institutions working with youth (government offices, churches, non-governmental organizations, HIV comprehensive care centers, compassion homes that care for disadvantaged children, and schools) [15].

Rennie et al. (2018) [59] explored community members through CAB and the youth advisory boards involved in recruiting study participants by consulting staff of institutions working with youth. Before the study implementation, Tarr-Attia et al. (2018) [71] explored community members through CAB and the youth advisory boards involved in recruiting study participants by consulting staff of institutions working with youth. Before the study implementation, Simwinga et al. (2016) [62] explored community engagement for HIV combination prevention therapy. Before the study's implementation, 'Broad Brush Surveys' (BBS) were conducted to understand the communities. The findings, combined with iterative discussions between the social science, intervention, and CE teams, informed the development of strategies for community representation. The suggestions provided by the CAB were also recognized and used for protocol and tool revision and intervention message development [62].

In a study by Davies et al. (2012) [66], community members participated in designing user-friendly promotional event reporting forms to capture information on events associated with ACTs. Strategies were developed for participant engagement and pretesting phases [66]. Freudenthal et al. (2006) [46] reported the community engagement activities/school activities in the screening and treating schoolchildren for schistosomiasis and intestinal helminths. The school activities include the development of slogans for prevention, writing school essays, preparing video-recorded dramas, songs, and dances, and household sanitation surveys for the students. The pupils were involved in the participatory action research as researchers and change agents [46].

Mtove et al. (2018) [16] described the multiple-level stakeholder engagement in malaria clinical trials evaluating IPTp. The study indicated that the level of engagement was at different levels with multiple stakeholders, including the local community. At the international level, various stakeholders were engaged. They influenced the study design, representing the community members, who were engaged and influenced the study design and research ethics. At the national and district levels, stakeholders such as government departments, ministries of

health, and ethics committees were engaged and influenced by regulatory requirements, standards of care, and research ethics [16]. Community members participated in designing effective interventions for controlling urinary schistosomiasis through participatory action research [58].

A study conducted by Angwenyi et al. (2020) [17] explored the basic and formal community engagement activities that took place over the first few years of the paediatric randomized controlled malaria vaccine trial conducted by the Kenya Medical Research Institute/Welcome Trust Research Program (KEMRI/WTRP) in three sites within Kilifi County, Kenya. Throughout the trial, the research team generally considered field staff and community leaders to be the channel for community members to voice concerns. In the trial, community health workers (CHWs) and field workers identified and recruited children. The consent processes for the trial were embedded into the broader community engagement activities but considered as a separate, more specific, and individualized activity [17].

A community-based participatory research approach by Mabunda et al. (2016) [50] for adapting TB-directly observed treatment intervention programs in South Africa explored the engagement activities in the program. The study indicated that the planning team, consisting of TB coordinators (provincial TB managers and district TB coordinators), was involved in recruiting participants and conducting focus groups [50].

A participatory action research by Olaseha and Sridhar (2005) [58] explored community engagement in controlling schistosomiasis. First, an intervention planning network was created to design and implement effective interventions. The Department of Health Promotion and Education of the University of Ibadan, with the students and specialist staff, took leadership while other agencies involved carried out supportive roles. The parents and teachers of the affected pupils were contacted and adequately informed at their various PTA meetings. A Ward Health Council (WHC) was formed to represent communities and work and collaborate with the created network (team) [58].

Shahmanesh et al. (2021) [60] identified community-based participatory research to adapt the per-led intervention to support HIV prevention. The community leaders selected peer navigators from their community and underwent 20 weeks of training [60].

*4.1.3. Community engagement during the conduct of research.* A relatively large number of included studies (27/40) reflected how communities engaged in implementing the research plans, including activities such as data collection, community entry or study initiation and sensitization to build trust, supervision, monitoring, quality control, and implementation of interventions [11,13–19,43,44,46,47,49–52,54–56,58,60,62,63,65,66,71,73].

In 12 studies, the community members were involved during community entry and sensitization to promote visibility and local ownership [13,15–19,43,51,56,58,63,73]. Nyika et al. (2010) [13] met with community members, including heads of families, to inform, sensitize and invite potential participants. A group of students and staff of the Department of Health Promotion and Education of the University of Ibadan were held to sensitize the communities and teachers about the nature and the challenges of the problems in participatory action research by Olaseha and Sridhar (2005) [58]. The involvement of mass media worked to sensitize residents about the outbreak of schistosomiasis in the study communities as part of the Information Education Communication (IEC) [58]. Pare Toe et al. (2021) [18] described engagement activities relevant to field trials on non-gen drive genetically modified mosquitoes. At the village level, the understanding was gained from ethnographic studies that highlighted "gate-keepers" in the community, official and non-official power structures, dominant and minority social groups, and the quality of the relationship between village leaders and administrative authorities [18].

A study by Angwenyi et al. (2014) [17] explored the community engagement activities of pediatric randomized controlled malaria vaccine trials in Kenya. The trial emphasized

consultation and sensitization with key stakeholders, including the provincial Administration (district officers, chiefs, and village elders) and the MoH (district health managers, the local health facility staff, and community health workers) at the outset. At each site, initial consultations (community entries) with community leaders (community gatekeepers) were followed by sensitization meetings for the general community, mainly targeting families with eligible children. Meetings were organized by trial staff in liaison with community leaders, with key messages developed by the Community Advisory for Study Teams (CAST) group. Although the overall community sensitization plan for key stakeholders was the same across sites, some changes were reported and observed over time. For example, community health workers (CHWs) were increasingly involved in giving general information about the trial to potential participants in the community. However, over time, more community-based information sharing is needed.

Trial staff attributed these changes to less need for large-scale community-based information over time because of fewer concerns about the purpose, amount, and use of blood taken and about trial safety. Overall, trial staff, community leaders, and participating households reported that community engagement activities and interactions helped clear pre-existing concerns and misconceptions about KEMRI's work; raised awareness; built trust; and increased the visibility of KEMRI staff initially seen as outsiders.

"*It was very, very hard, in fact, at the initial stages . . . KEMRI was being associated with devils, and that was the story in the community then, but I commend the entry point. Before they started this whole project, there was very, very good interaction between you people [KEMRI staff] and how you entered these communities through DHCs [dispensary health committees], chiefs, and barazas [public meetings]. It helped a lot. I think that's why, after a short time, community members accepted because now, I don't hear any problems or maybe community members complaining. . .*" [Group interview, health facility staff]

A study by Bandewar et al. (2010) identified three distinct phases of community engagement in the Majengo Observational Cohort Study (MOCS) that examined sexually transmitted infections, in particular, HIV/AIDS [65]. The MOCS needed a plan and preparation for community engagement at the beginning. However, the need for specific strategies and practices for community engagement evolved gradually by the cohort study researchers. Communities were engaged in (a) reaching out: mobilization, dialogue, and education; (b) foundations of trust through relationships of care; and (c) leveraging existing 'social capital' to form a cohort community. Engaging sex workers in Majengo village required a range of skills and strategies. The cohort study researchers recognized, in particular, the need for an experienced person from the field of community health:

"*. . .they [the Project team members] realized that I could mobilize the community, I could get them [women sex workers] together, I could communicate the essence of health-seeking behaviour that is of benefit to them. And so, they asked me to team up with them to mobilize the women to visit the Majengo clinic*" [Community health expert]

The first formal engagement strategy adopted by the research team in MOCS was a 'door-to-door' outreach in the Majengo village and 'one-on-one communication' with the sex workers as described by one interviewee:

"*. . . So, it was very challenging to go from door to door trying to figure out how to establish rapport and open a dialogue with a woman. And it is not easy for someone to say that, "Yes, I do this [sex work]." But they eventually did come on board*" [Community health expert]

In a study by Davies et al. (2012) [66], community members reviewed the existing forms, developed a new form, and pretested a user-friendly adverse event reporting form. The existing forms were examined by two focus group discussions with community medicine distributors (CMDs) and health workers, in which the participants were encouraged to discuss their experiences with anti-malarial treatment and reporting of adverse events with existing forms and procedures. This helped them understand how Uganda's current pharmacovigilance reporting forms were used. Following the review, two participatory workshops were held, one with CMDs and one with health workers, with the involvement of local artists and the production of draft forms. The local artist drafted and refined sketches at the request of participants and practiced in the field [66].

A study by Dierickx et al. (2018) [19] identified the relevance of community sensitization for individual decision-making in a clinical trial on malaria carried out by the Medical Research Council Unit Gambia. The study revealed that community sensitization effectively provides first-hand, reliable information to communities as the information is cascaded to those who could not attend the sessions. The information given during the community sensitization and individual informed consent process was consistent with the information written in the consent form and information sheet [19].

Another study by Diallo et al. (2005) [43] describes obtaining community permission at Mali's malaria vaccine study site. The study identified the Malian experience of the community permission process that had six steps: a) a study of the community (to elucidate the community's socio-cultural structure and health system and identify political and traditional leadership); b) an introductory meeting with leaders (to introduce the research team, begin explaining the research project, and solicit the best process for community permission); c) formal meetings with leaders (to explain the research project, risks, benefits, etc., in detail and to take and respond to questions); d) personal visits with leaders (to visit leaders personally in their homes for further explanation and opportunity to answer their questions); e) meetings with traditional health practitioners (to develop a formal agreement with traditional health care providers for collaboration on the research project); and f) recognition that obtaining permission is a dynamic process (to conduct a modified consultation process with leaders at each modification in the protocol or new research project) [43].

A study by Mtove et al. (2018) [16] described stakeholder engagement in malaria clinical trials evaluating IPTp. The study indicated that stakeholders, such as the community, family, and study participants, participated at the local community level. Community engagement measures undertaken by investigators included local meetings with community leaders to explain the research aims and answer questions and concerns voiced by the community (consent, perception, values, and customs) [16].

Nakalega et al. (2021) [55] explored community engagement on ethical considerations for involving adolescent girls. Adolescent girls and young women attended a stakeholder consultative meeting to create awareness about the study, gain community insights, and learn about community perspectives on engaging adolescents younger than 18 (the legal age of consent) in sexual reproductive health-related research.

Marsh et al. (2011) [51] explored the role of community engagement in international collaborative biomedical research. The engagement activities include: a) public engagements through schools, public meetings and events in the community, regular interactions with opinion leaders who are in continuous contact with the broader community (to strengthen the general awareness and understanding of biomedical research concepts and activities); b) public meetings and participatory workshops that support the interactivity between researchers and the community (to promote visibility, accountability, reliability and perceived fairness were conducted to build an appropriate level of trust); c) meetings with community representatives

i.e., community members and opinion leaders (to support consultation/ deliberative discussions and to understand how general or specific research project activities may interfere with freedom of choice); small scale meetings with invited groups including field workers and drivers (to build trust on specific studies); and meetings/discussions with 'typical' representatives and opinion leaders (to understand how research activities may generate 'hidden' costs or benefits, and to ensure the validity of science).

Nakibinge et al. (2009) [56] explored community engagement in health research from a project on HIV in rural Uganda. The study indicated that a community project advisory board initially played a key role as a community liaison. The community consultation and feedback were subsequently assured through the local council (LC) system of civic administration, thereby facilitating sustainability and community acceptance. The project engaged the formal LC community leaders and informal leaders (individuals with considerable influence on community opinion by their status or reputation). An influential informal leader was recruited as the first community liaison officer, and other informal leaders were recruited as advisers and guides during annual survey rounds [56].

Participatory action research by Freudenthal et al. (2006) [46] indicated that the data generated from the screening for schistosomiasis and intestinal helminths, school essays, dramas, and household sanitation surveys were used during feedback meetings in schools and the broader community. Reflection on the household sanitation survey was part of the school activity. Based on the reflections, different prevention mechanisms were introduced by schools and community members [46]. Community engagement through schools, public meetings and events, and regular interactions with opinion leaders in continuous contact with the broader community were done to strengthen the general awareness and understanding of biomedical research concepts and activities to build trust in specific studies [51].

Meiring et al. (2019) [52] reported different meetings with community representatives, community leaders, individual schools, and community health committees held to present a study overview, seek the participation of the community in the vaccine trials, and establish feedback from the community. Additionally, the study reported that a mobile Van with audio recording, which a local musician prepared, was used to inform people and invite them to vaccine clinics. The local musician created a jingle containing study messages and invited guardians to bring children for vaccination [55]. *Mabaraza*, a traditional community assembly used for information sharing and gathering community opinions on issues, was used to engage community members during the community entry and consent process [63].

A study by Shahmanesh et al. (2021) [60] identified community-based participatory research to iteratively co-create and contextually adapt the per-led intervention to support HIV prevention. The peer navigators discuss the vignettes and co-create the Thetha Nami ('talk to me'). This intervention included peer-led health promotion to improve self-efficacy and demand for HIV prevention, referrals to social and educational resources, and accessible youth-friendly clinical services to improve uptake of HIV prevention [60].

Study investigators undertook community engagement measures with community leaders to explain the research aims and answer questions and concerns voiced by the community (consent, perception, values, and customs) [16]. The study also engaged with family members of prospective trial participants to be sensitive to local practices and beliefs [16]. Adolescent girls and young women attended stakeholder consultative meetings to create awareness about the study, gain community insights, and learn about community perspectives on engaging adolescents younger than 18 (the legal age of consent) in sexual reproductive health-related research [55]. Community consultation and feedback were subsequently assured through the local council (LC) system of civic administration, thereby facilitating sustainability and community acceptance for the research project on HIV epidemiology [56].

Community members were engaged in data collection activities [14,50]. For instance, Doshi et al. (2017) [14] described community researchers who are members of the MSM played a central role in collecting data in a study that investigates the willingness of MSM to participate in HIV vaccine efficacy trials. The community researchers collected narrative information directly from the MSM through individual in-depth interviews (IDIs) using semi-structured interview guides containing mainly open-ended questions and some prompts [14]. Community representatives conducted focus group discussions in a community-based participatory research approach to needs assessment for adapting TB directly observed treatment (DOT) intervention program [50].

Communities engaged in the implementation/intervention of the research activities were also mentioned in 15 studies [11,15–18,44,47,49,50,52,54,56,60,62,71]. A community-based participatory research undertaken by Faye and Lugand (2021) [44] described the development of information, education, and communication tools to promote intermittent preventive treatment of malaria in pregnancy (IPTp) in the Democratic Republic of the Congo, Nigeria, and Mozambique. The researchers interacted directly with community participants on the field by listening to them and collecting their opinions on improving tools. The graphics used in IPTp communications tools were modified according to the respondents' feedback to make them more culturally and socially sensitive and validated [44]. In a technology 'co-development' that involves collaborative partners in Africa, Europe, and North America, the Malian engagement team was responsible for working with stakeholders at the local, national, and regional levels and undertaking community engagement around its locality and sharing short videos and materials about its activities [47].

A study by Broder et al. (2020) [11] described members of the AMP community working groups participating in monitoring emerging issues. Community working groups consisting of community educators were also responsible for assuring that CAB representatives have input into study-specific topics such as addressing community misconceptions, informing researchers of local issues or concerns that can affect the conduct and successful implementation in that locale, determining appropriate and non-coercive incentives for trial participation and to support retention, as well as determining the package of services that make up the local standard of HIV prevention [11]. Community members were engaged in participant follow-up and retention and ongoing community feedback to and from the community [17,49,54]. Community members or end-users participated in workshops to help solve problems and design user-friendly advert event reporting forms [66].

*4.1.4. Community engagement in data analysis and interpretation.* Only seven studies (7/40) engaged communities in data analysis and interpretation, mostly verifying and giving feedback on the results [17,46,49,60,66,72,73].

In the MSM study, [14] community researchers who were selected from their respective MSM communities in a study that contextualized the willingness of MSM to participate in HIV vaccine efficacy trials participated in the data analysis of qualitative interviews. A participatory action research [46] recognized that community members jointly analyzed the data generated. They played a more significant role in identifying what development activities could be undertaken in these contexts to reduce transmission.

Community members engaged in providing input to the study results or interpreting the study findings [11,17,46,49,60,66,72,73]. A study by Agot et al. (2019) [72] involved former study participants to give them context and interpret the study results. The former participants of the TRIO study (that assesses the preferences for attributes of tablets, vaginal rings, and injectable products for dual prevention of HIV) attended different dissemination and one-on-one sessions. Over 40% of former participants participated in the sessions six months after

exiting the study and highlighted their keen interest in learning the study results and having an opportunity to discuss them with the study team.

A study by Shahmanesh et al. (2021) [60] explored community-based participatory research to co-create a peer-led intervention for HIV prevention. Based on the feedback, the peer navigators refined the Thetha Nami intervention to include other components [60].

Community members giving feedback on the preliminary results helped interpret the findings [17,49,73]. Beard et al. (2020) [73] described the feedback mechanisms for research findings that involved community members. According to Beard, a conference advisory was initiated, and after three rounds of data collection, preliminary research findings were presented to different stakeholders, including the study participants and the general community. Another study by Kamanda et al. (2013) [49] showed that community advisory boards (CAB) had quarterly meetings with the research team to review preliminary findings, discuss interpretation within the local context, and determine appropriate dissemination strategies. Davies et al. (2012) [66] explored that end-users were involved in testing the tool or form with adverse event scenarios for ACTs. End-users give feedback on the tool, and tool redesign is performed accordingly.

Community working groups increase community members' capacity by participating in regional workshops and giving context to the results [11]. Students conducted a household sanitation survey and presented the results for feedback during school and larger community meetings [46].

A study by Angwenyi et al. (2014) [17] revealed feedback to the study communities on preliminary results. The meetings provided an opportunity to reiterate the study aims and importance of the trial and thank participants' parents for participating. The findings presented to the participating families, community leaders, and local health facility staff appeared to alleviate further concerns about the trial and encourage participating parents to keep their children in it.

*"Mothers [parents] were very happy because at least they saw that their efforts of coming here [at the health facility] for the vaccine was not wasted and so there was something truly going on. Therefore, they knew this study was real when they got these results. Another thing they were happy about was the way they were being followed up by those [fieldworkers] who oversee the project at their village. It was a clear indication that they [parents] were a very important link in the study"* [Fieldworker in Site A]

*4.1.5. Community engagement in dissemination and translations of the research findings.* Eight studies engaged communities in disseminating the research findings through various mechanisms such as giving/attending presentations at meetings and conferences and translating research findings into policy products such as policy briefs [11,17,46,49,60,66,72,73].

A study by Agot et al. (2019) [72] involved former study participants attending dissemination and one-on-one sharing sessions six months after the study. The exercise helped them to ensure that correct lessons were derived from those results and increased the credibility of the findings reported by the investigators [72].

Communities engaged in the dissemination workshop of preliminary study results [11,17,49,73]. Community members helped the research team produce and distribute advocacy communications facilitating behaviour change [73]. Kamanda et al. (2013) [49] explored community engagement activities while disseminating results. The study indicated that dissemination of study results is done locally (to the local community via mabaraza, media, and Children's Services Forum) and internationally (to the international community via conferences and journals). Presentations were done in lay format to disseminate information directly

to the community to gain valuable feedback and ensure ongoing cultural and community relevance [49].

Community members were engaged in translating the study findings into action [46,60,66,73]. A community-based participatory action research by Freudenthal et al. (2006) [46] reported that community members decided to create safe swimming places to prevent schistosomiasis, and teachers developed a curriculum for schistosomiasis education in primary school. Peer navigators who were community representatives co-created and contextually adapted the biosocial peer-led intervention to support HIV prevention [60].

According to a study by Beard et al. (2020), research teams and community members produce and distribute advocacy communications to facilitate behaviour change at the district level. According to Beard et al., a film entitled "Keeping the Promises" highlights the challenges community members face when attempting to access health and government services. Three local newspaper articles and two radio programs feature the film and the need for an integrated child welfare management plan. Davies et al. (2012) [66] reported that a participatory approach was used to create novel, effective, and user-friendly reporting forms or tools that non-clinicians can use to collect much-needed pharmacovigilance data related to adverse events associated with ACTs.

**4.2. Level of community engagement in research.** Taking the WHO community engagement framework as a conceptual basis, all included research projects passed through the lowest level of community engagement, i.e., *informed* [11–19,43–52,54–56,58–67,71–73]; 31 studies engaged communities at *consulted* stage [11–18,43–46,48–52,55,56,58–64,66,67,71–73]; 10 studies at *involved* stage [11,14,46,49,50,58,60,64,66,73]; five studies *collaborated* with communities [11,46,49,60,66] and only one study by Freudenthal et al. (2006) [46] *empowered* the communities in the research process.

In summary, as indicated in Fig 2 below, regarding the community engagement related to the stages of the research process, the highest score was documented in the research implementation and design stage, while most engagement levels were at informing and consulting levels.

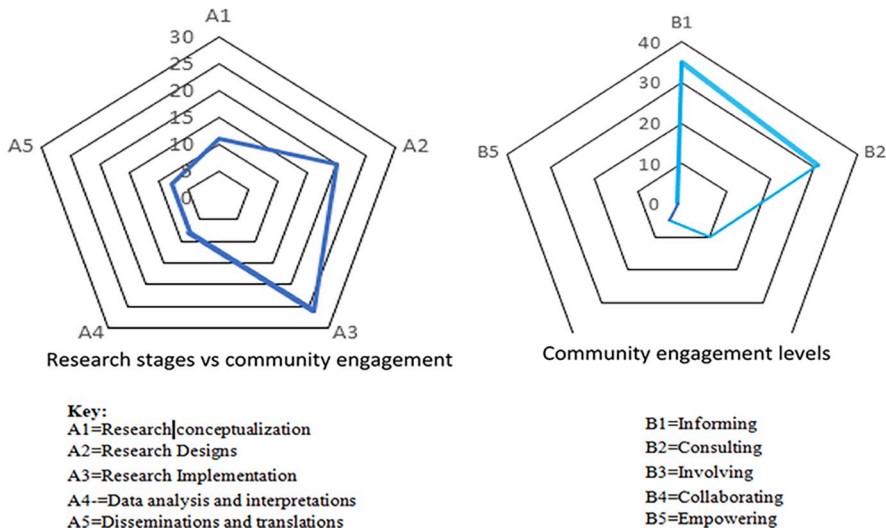

**Key:**
A1=Research conceptualization
A2=Research Designs
A3=Research Implementation
A4-=Data analysis and interpretations
A5=Disseminations and translations

B1=Informing
B2=Consulting
B3=Involving
B4=Collaborating
B5=Empowering

**Fig 2. Score of community engagement by research stage and level of engagement.**

**4.3. Challenges and mitigation strategies of community engagement in research.** Eighteen research reports [11,13,14,16–19,49,51–53,56–58,60,62,64,73] mentioned the challenges encountered and their mitigation strategies in engaging communities in the research process (see S4 Table for details). These challenges are related to poor consenting and ethical dilemmas, communities' perception and inadequate knowledge of research, poor representativeness and partiality of advisory boards, lengthy research process and high communities' expectations, participation fatigue, lack of shared control over the research by the community, late communication of study results, lack of funds for sustainable community engagement in research, and other challenges related to a multinational research project.

*4.3.1. Ethical dilemmas and consenting.* In some culturally sensitive research projects, the ethical issues in engaging vulnerable populations were found to be a challenge. For example, a Kenyan study involving Men having Sex with Men (MSM) [53] identified that working with different Men having Sex with Men risks inappropriate labelling. It also created false categories that do not resonate with reality or that benefit some at the expense of others, possibly raising stigma and discrimination. Indeed, mitigation strategies such as honest and open communication about the goals of engagement, ensuring that what is promised and given to the group and communities as part of research is not organized in such a way as to undermine individuals' abilities to make free and informed choices about whether to participate in that research; avoiding internal conflicts within and between different groups; exploring and considering the views and priorities of the most vulnerable group; explaining to the general population for working with such groups, and delivering carefully worded key messages to these groups was to minimize negative stereotyping and misconceptions that reinforce stigma [53].

On the other hand, both practical and ethical challenges were encountered during consenting and assenting procedures [57]. For example, some community leaders attempted to pressure people to enrol and the community's misunderstanding of the message on the information sheet was challenged in the consent or assent process [16]. From a practical point of view, despite community engagement activities and community leaders sharing information and answering questions related to consent and ascent, there were still misconceptions within the community [52]. Improving communication and understanding of the research and maintaining dialogue with all the relevant stakeholders were also used in solving such problems [57].

One of the research projects has also overcome the adverse effects through individual-based information sharing and conducting the consent process by trained clinical staff [17]. Providing information and answering questions at enrolment was also crucial for addressing misconceptions and helping individuals make informed decisions. Information provided at enrolment travels through the broader community because participants deliver those messages back to friends and family [52].

To address the challenge of navigating language and literacy issues in obtaining informed consent, consenting participants in their preferred language and confirming an acceptable literacy level was utilized [16].

The ethical dilemma has several risks, including perverse outcomes related to existing social relations in communities and conditions of 'half knowing' intrinsic to processes of developing new understandings, which is critical. A broad approach to building mutual understanding and trust between researchers and community members is essential in addressing ethical issues and dilemmas [51].

*4.3.2. Communities' perceptions and knowledge of research.* Few studies identified factors influencing a community's willingness to participate in the research [14,60]. These studies suggested strategies towards community engagement and recruitment of trial volunteers

informed by analysis and understanding of individual/socio-cultural motivators/barriers, acceptability, and experience with/knowledge of medical research [14,60].

Notably, researchers considered that not all cultural values equally recognized by different age groups of the same community; for example, in a study by Shahmanesh et al. (2021) in rural South Africa, Thetha Nami 'talk to me' is not being valued by youth, seen as "useless by youth." The team navigators solved this challenge by deconstructing what these challenges signified. After doing so, they identified it as partly due to the logistic barriers they faced in delivering Thetha Nami: small age differences, particularly with older adolescents; lack of safe spaces for youth to gather; and size of the rural areas with poor transport links. On reflection, they identified the "uselessness" as signifying gaps in the services they had at their disposal to offer youth [60].

In one study [62], the Community Advisory Board (CAB) complained that their community group was inequitably treated because of their assignment to the control arm of a trial. According to this view, communities in the control arms needed to be better informed about what was happening in the study and felt the approach needed to be more respectful. In response to this challenge, the project created a separate advisory structure, called the Community Partners Platform (CPP), to bring the civil society perspective into the consultations about study implementation, with a secretariat hosted by the Treatment Advocacy and Literacy Campaign (TALC). After this consultation, additional field staff was employed to facilitate engagement in the control arm. CAB meetings also became more frequent, monthly instead of quarterly. Study sensitization activities were introduced with care taken to ensure that messaging was relevant to control community activities. CAB members visited the study laboratories and were taken through the receiving, processing, and storage procedures [62].

*4.3.3. Poor representativeness and partiality of advisory boards.* Using an advisory board is considered to engage the community through their representatives, but poor representativeness and partiality of the advisory board were found to be challenging [18]. Using locally sensitive and recruiting advisory board members through community consultations helped avoid such challenges. Applying this required assessing the power dynamics and relationships between different social levels and how this could impact the decision-making processes upon being members of the community advisory board. The research team conducted an informal assessment by collecting stakeholder feedback and conversations between the engagement team and community members [18].

*4.3.4. Lengthy research process and high community expectations.* Potential challenges such as lengthy research processes and the community's expectations that may be too high to meet were challenges identified in some research projects [13,49]. Eliciting the community's feedback through multiple sources was also found to be a challenge. Their need does not balance with the scope of research projects. To address such challenges, one of the research projects has taken the time to listen to and sympathize with the communities' concerns. The research team has understood open and ongoing communication, understanding, and Community Health Workers who act as the direct link between the project and the community have been fundamental in balancing community needs and requests with the scope and aims of a research project [13,49]. Most importantly, community engagement by itself was found to be an appropriate platform to address part of such challenges [13].

*4.3.5. Participation fatigue.* Researchers can employ several strategies to combat participation fatigue and maintain community interest and motivation. This includes offering appropriate incentives and ensuring they align with community needs and are culturally sensitive. Additionally, providing high-quality services through effective communication, timely provision of services, and prompt issue resolution builds trust and strengthens the relationship between the research team and the community. Actively involving community members in

the research design phase fosters a sense of ownership and investment. Introducing innovative research approaches and methodologies sparks curiosity and engagement. Regular communication, sharing progress updates, and seeking feedback demonstrate transparency and keep the community informed. Tailoring engagement strategies to diverse subgroups within the community acknowledges their unique needs and preferences [55]. Moreover, continuously assessing the research's relevance to the community's priorities and needs ensures sustained interest. Implementing these strategies cultivates a positive, enduring partnership between researchers and the community, ultimately enhancing the quality and impact of the research endeavours".

*4.3.6. Lack of shared control by the community*. Lack of shared control by the community could be a challenge during the implementation of the research. It may create tensions between the research team and the community. Careful characterizing the community and its changing needs, establishing trust with stakeholders, and developing community assets could help overcome these challenges in early initiating engagement activities [73].

*4.3.7. Lack of funds and shared priorities for sustainable community engagement*. Lack of funds to implement and sustain planned community engagement activities of a program was one of the challenges encountered [58]. The community's needs are not always in line with funding agencies' agenda; communities and Community-Based Organizations (CBOs) are interested more in services than research, while academics are interested more in research than services, making it hard to build a sustainable partnership between communities and researchers. These challenges were addressed by expanding the interests and activities of academics beyond the campus environment, offering the opportunity to research the integration of theory and practice [58].

*4.3.8. Late communication of study results*. Late communication of study results in an audience-appropriate manner was challenging in one research work. The research team solved this challenge by analyzing and publishing secondary data through reports using white papers [73].

## Discussions

This systematic review analyzed existing evidence, practices, and lessons regarding community engagement in research addressing health care delivery in infectious diseases of poverty in SSA. Accordingly, we conceptualized community engagement in research according to a classical component of the research process, namely, conceptualization of the research; designing research strategy, methods, and tools; implementation or carrying out of the research; data analysis and interpretation; and dissemination and translation of the findings [34]. To create a genuine community engagement opportunity, it must be initiated as early in the research process as possible. Identifying and conceptualizing the needs and gaps within the target community is vital. This helps establish solid foundations for ensuring community needs, aspirations, and knowledge are used to inform further development of the research techniques. Unfortunately, our review indicated that community engagement at this critical stage of the research, where research ideas are born and have a shape to evolve, is reported only in a few studies.

The classical approach, whereby the researcher could identify research questions, needs, and gaps from one's perspective, dominates the creation and formulation of the research idea and its conceptualization process. Thus, it is logical to argue that the research ideas generated in such an expert-dominated strategy may substantially lack local relevance and trust. Moreover, collaborative and mutual learning for the co-creation of knowledge could be missed.

The few studies included in this review attempted to engage communities at the early stage (framing research questions and conceptualization)- where communities contributed their input and feedback to identify research problems and set scientific agendas [11,58,62,66,71].

The participation of the communities was assumed indirectly through engaging or establishing community engagement platforms or structures like community advisory boards, community working groups, community representatives, site advisory boards, community leaders, and representatives [11,58,62,66,71]. The assumption was that these community leaders and representatives could initiate communities and serve as a voice for research communities. It also helps to define or redefine the research issues/agenda and ensure the cultural acceptability of the research methods and procedures at the outset. By doing so, active participation and a sense of ownership in the community could be facilitated [17,46,55,66].

Formal and informal face-to-face discussions between participants and stakeholders are essential for developing research concepts. These include community visits, storyboards, diary tables, and field notes. Meeting with community representatives, listing priority health problems, and setting research agendas and priorities are also helpful in engaging communities at the conceptualization stage [45,58,66,67]. Depending on the context, using technological and digital platforms such as tele/video conferences and calls could also potentially engage communities during problem identifications and conceptualizations [11,12,46].

This review depicted limited available evidence and provided pieces regarding ways to engage communities in developing research design and strategy. These include input into critical decisions such as choosing research methods, developing sampling procedures, designing interviews and/or survey questions or any research tool or formats including consent forms and others [11,13–17,43,46,49,50,55,58–63,65,66,71,73]. Community advisory boards (CAB) and other relevant community-level actors (existing community groups and networks) substantially facilitated and assisted community input and feedback. This was done during the development of strategy and protocol by informing effective strategies for recruitment and retention of research participants, developing and adapting consent forms and study materials for local use, incorporating acceptable ethical approaches [11,55]' and participating in the design of workshops and capacity development for selected community members [11].

Depending on the local context, the CAB plays multiple roles [55], which are crucial for the success of the research, from ensuring that the research fits and addresses the local needs and views to building trust and partnerships were also crucial in shaping the research strategy and effective implementation [74]. Thus, research institutes and academics must ensure functional and representative CAB that fully and responsibly operates in the entire research cycle. Interestingly, community educators have the potential to educate communities about the research agenda and collaborate with the CAB at the field level, especially to reach out to marginalized and hard-to-reach populations geographically with unique cultural experiences [11].

Few experiences also indicated that community leaders and local administrators, community networks, youth groups, school students and teachers, and religious leaders could assist or complement the function of CBA by facilitating or serving as an agent to initiate and promote community participation at every cycle of the research [15,58,59,73]. Indeed, there were experiences where school students contributed to designing and developing research education materials and intervention messages such as poems, social dramas, songs, and dances, making them effective research agents [46,75]. Given that research design and strategy form the blueprint and roadmap for the entire research piece, the need for full-scale collaborative design with local stakeholders -and target communities is essential.

This review revealed that community engagement was more prevalent and prominent during the actual field implementation of the research. This suggested that community engagement is often initiated at the implementation stage of research with the diverse role of the communities ranging from using the community members as passive suppliers of information or passive users of the interventions (not considered as engagement in this review) to active engagement in the implementation process. These include facilitating community entry,

community initiations, and sensitizations to build implementation support, building trust, carrying out and assisting or supporting in data collection (e.g., community representative participates in the conduct of FGDs, student conducted household sanitation surveys). Besides, recruiting participants for the study, building trust and supervision and promoting the project for visibility; conducting research interventions (such as informing, educating, and mobilizing); monitoring adherence to intervention protocols and adverse events, and community reactions, and acceptance; providing feedback to adjust or modify the interventions protocols, study procedures and messages as appropriate were also some of the activities during the implementation process [11,13–19,43,44,46,47,49–52,54–56,58,60,62,63,65,66,71,73].

Relatively, more meaningful and diversified engagement strategies were captured for engaging communities in the implementation, including sensitization and introductory meetings with community members, mass media news and education, participatory workshops, and focus group discussions. These strategies helped to gain feedback on the implementation process, conducting community dialogue, working CAB for community liaison, consultations and personal visits to community leaders and gatekeepers; office and non-official power structures; regular and frequent interactions with the research participants; use of tailored IEC materials and short videos; door-to-door mobilization and one-to-one communications especially for sensitive issues traditional community assembly or vents village/opinion and community leaders [13,16–19,47,51,55,56,58,60,66].

Even though this review captured substantial lessons and positive practices that can promote community engagement in the execution of research activities, there is a need to go beyond the conventional approaches to innovative and context-specific approaches to achieve real community empowerment. This is because participation in executing the planned task offers an opportunity to build community capacity and knowledge through learning by doing. Indeed, practices such as engaging community representatives and volunteers to serve as researchers and citizens "science research" with appropriate technical support and facilitation by the researcher. This could help to diffuse scientific and research literacy in the general public [76].

Engaging research participants in the interpretations of the findings could help ensure the findings are credible, meaningful, and reflective of reality as a broader societal agenda and minimize researchers' influence or bias on the study. This allows the study community to reiterate the study's aims and importance and is part of acknowledging and giving credit to the research participants [17]. In our review, only a few studies [17,46,49,60,66,72,73] reported community engagement in analysing and interpreting the results. This was done through different methods such as engaging community representatives in the joint analysis [46]; sharing the preliminary results with community representatives, study participants, and stakeholders for input and feedback [11,17,46,49,60,66,72,73]; inclusion of research participants at research conference; dissemination and validation workshops, and holding specific meetings [17,60,72].

Like in other research stages, the CAB plays an active role in data analysis and interpretations by ensuring an accurate interpretation of the findings according to the local context and credibility and validity from the perspectives of the study community. This can be done by letting the CAB review the preliminary findings and provide guidance on interpretation within the local context [49]. To maximize the potential role of CAB and community leaders, developing and enforcing an explicit mechanism for preliminary data sharing with the community advisory groups is essential. Researchers should also have a clear strategy and plan for engaging communities in the analysis and interpretation of the findings.

To ensure the research findings are effectively translated into policy and practices, target communities and local stakeholders should actively and interactively engage in the process.

This is essential to ensure that the policy products that emerge from the findings are translated to ensure the policy recommendations' meaningfulness, appropriateness, and acceptability. Thus, instead of relying on the passive transfer of information, researchers should collaboratively and collectively identify, filter, interpret, adapt, contextualize, and communicate the evidence for policymaking. This requires building an effective strategy to engage communities in disseminating and translating research outcomes from the outset of the research. Unfortunately, the depth and breadth of experiences regarding engaging communities in translating the findings into policy and practices are so limited.

Researchers often relied on conventional methods of research output dissemination through scientific meetings, conferences, dissemination workshops, presentations, and publications to share their findings, which gives little opportunity for the community to engage [7,13,49,52,69,75,76]. Even when translations are made (e.g., producing advocacy communications, policy briefs, newspapers, newsletters, short films, or videography), the process is dominated by research with little or no input from the research community [11,17,49,73]. A study by Freudenthal et al. (2006) [46] enabled the study community (students and school teachers) to translate the findings into actions by developing safe swimming places to prevent schistosomiasis and developing a curriculum for schistosomiasis education in primary schools, where the educational materials collaboratively created by communities and school students based on the findings of the study [60]. Other techniques include engaging communities and local stakeholders in defining policy advocacy, communicating the findings, and framing content for local newspaper articles and radio programs [66].

This review provided helpful insight into the need to promote and move away from expert-dominated research tools, methodologies, and techniques toward participatory, collaborative, and mutual learning processes [46,48,49,58], locating community as research partners in the entire research process [50,71]. In light of this, co-development strategies Delphi techniques [45,64,67], community dialogue, and working community leaders [49,64] by offering an opportunity to craft and develop research ideas, design protocols, and tools, and implement the research strategy are crucial to engaging communities across the spectrum of the research element [46,47]. However, applying participatory research requires researchers to be open-minded with a sense of democratic accountability and receptive to various ideas and new experiences [47,77,78].

Although participatory research approaches and co-development frameworks may not suit every context, fields like public health, socio-behavioural studies, and, most importantly, qualitative researchers and researchers would benefit from it [47]. Participatory research approaches require greater skills and knowledge of how to plan, conduct, and manage it, and thus, health research training (curricula, short courses, and hands-on training) and capacity development activities should adequately incorporate the concepts, methodologies, and tools participatory in addressing practical and theoretical aspects.

In this review, we recognized that complete and comprehensive evidence and best practices with greater depth and breadth regarding community engagement in research is generally lacking. There is a limited spectrum of community engagement with low levels of participation, namely informing and consulting, thereby leaving communities without empowerment. Only some studies achieved full-scale community engagement in the research process. Some initiated the engagement at the design phase, while most concentrated on implementation., There were few instances of engagement during the analysis and nearly none in translations.

Additionally, the review identified significant barriers that potentially limited effective engagement. These include issues such as poor representativeness and partiality of the advisory board [18], absence of trust in the local health system [16], lack of mutual understanding and trust between researchers and community members [51,73]. Researchers' inability and lack of

skills and knowledge regarding planning and implementing participatory methods [64] also paused a challenge. Moreover, insufficient funds allocated to community engagement may be attributed to the researcher's oversight or result from inadequate attention by the funding agencies.

The other key challenge affecting community engagement is inherently different interests between and within communities and researchers where the communities are often interested in receiving services. In contrast, researchers and academics are interested more in research [58]. Addressing this challenge requires comprehensive and multilayer approaches such as increasing public awareness and broader understanding of the research; integrating open and ongoing feedback collection and analysis and response system; increasing researchers' and academicians' engagement with local communities and local stakeholders offering the opportunity for research for the integration of theory and practice [58]. Indeed, the community's expectation of obtaining services in research is a valid concern, and researchers and research stakeholders ought to integrate relevant packages of services and incentives in the form of health benefits into their research plan, as appropriate. This could help to reduce community fatigue, especially in areas frequently hosting research projects [55].

Most importantly, understanding the target communities' sociocultural motivations, influences, and power dynamics [14,60] and establishing community advisory boards with the effective linkage between the community and the research team/institutes is fundamental to balancing community needs and expectations [13,49]. However, caution must be made while using the community leaders and CAB as they could exert undue influence on the local community in favour of participation in the research under consideration, especially during the consenting and assenting process [57]. Thus, whenever community leaders, CABs, or influential community members participate in the research, it is essential to provide effective training and capacity building to minimize such pressure [16]. Limited general literacy in general and research literacy, combined with language barriers, is a persistent challenge [16]. Culturally sensitive research topics like MSM [53] may require careful consideration of the local norms and acceptable ways of introducing the subjects to avoid unnecessary risks (such as stigma and discrimination) to the study participants and adverse reactions from the community members.

### Limitations of the review

Although we conducted a comprehensive literature search from different databases and grey literature to capture unpublished studies or reports, there might be a possibility of missing some research studies. The review considered all study designs (quantitative, qualitative, and mixed methods studies); however, due to the limited availability of quantitative studies relevant to the objective of our review, we adapted our approach to qualitative synthesis methods. Thus, we might have missed out on some experiences reported quantitatively.

### Conclusions

This review synthesized community engagement practices and experiences in the research process addressing the infectious disease of poverty in SSA. It uncovered the scope, depth, and breadth of community engagement in the research process, mainly limited to informing and consulting forms of participation, with little opportunity for empowerment. To ensure effective and genuine community engagement in the research, initiating involvement as early as possible is crucial. Communities and relevant stakeholders should actively participate in the initiation, conceptualization, and design of research methods and tools, as well as the implementation, dissemination, and translation of the findings. Nevertheless, many research studies

and projects tended to begin their research study without involving communities. Many of them tended to initiate community engagement during the implementation stage of the research process. Community engagement in the conceptualization of the research project, analysis, dissemination, and interpretation of the result was rare. Most research projects only involved the community at a low level (i.e., informing or consulting them at some point) and showed the need to integrate communities throughout the research cycle.

Even though some details of how communities are engaged in the research process are lacking, valuable insights and lessons are drawn from the review. This includes the importance of establishing trust and ownership through understanding the values, norms, and culture of the community; exploring and harnessing community structure and platforms (such as community advisory board, community groups, and networks) for productive and successful engagement; being respectful to the culture and values, and norms of the community with adequately addressing diversities and adapting research methods and tools to community's context and culture. Moreover, researchers must move away from researcher-dominated tools and methodologies towards participatory research techniques, creating a broader understanding through effective communication in every stage of the research process, ongoing consultations and feedback, and ensuring researchers' integrity, commitment, transparency, and accountability through enforcing researcher obligation.

The review also identified pertinent challenges to community engagement that should be given attention in future community engagement efforts. These challenges were related to poor research literacy among the public and communities; inaccurate perceptions and high expectations from the research projects by the communities; limited funds for ensuring sustained community engagement in the research; researchers' lack of skills and motivations regarding participatory research process and strategies to effectively engage communities in the research; limited research communications and educations.

Despite community advisory boards and other community-based platforms and structures being excellent platforms to initiate and sustain community engagement in the research, the formation of such bodies might not be a true representative of the community, and adequate attention should be given to ensuring their representatives and fairness to be the voice of their community.

## Implications for practice

The following lessons learned are derived from careful analysis of the findings, which may inform researchers, policymakers in research institutes and universities, and other key stakeholders to promote effective and sustainable community engagement in the research process. In this regard, the following strategies could help to strengthen community engagement in the research process:

- Creating trust and ownership of the community through a careful understanding of the values, sociocultural system, power dynamics, and circle of influences

- Addressing concerns and misconceptions related to research in the community

- Exploring and harnessing community structure, forums, and platforms for productive and effective engagement

- Following culturally appropriate and acceptable recruitment strategies that respect the values and norms and are tailored to particular diversities

- Moving away from expert-dominated tools, methodologies, and techniques toward participatory research approaches

- Educating the public on research for a broader understanding of effective communication built in every stage of the research process

- Ongoing consultative community meetings, feedback sessions and forums

- Keeping integrity, longevity, and joint commitment between researchers and communities

- Ensuring transparency and accountability through enforcing researcher obligations and ethical standards

In addition to involving community members, engaging a wide range of stakeholders at every stage of the research process is crucial, from the project's inception to translating findings into actionable policies. This includes policymakers, healthcare professionals, advocacy groups, and other pertinent parties. By involving diverse stakeholders, research efforts can be enriched with a broader perspective and ensure that outcomes are relevant and applicable to a wider audience. Furthermore, the practical challenges faced and the mitigation strategies listed in this review could also help researchers plan their future research projects.

## Implications for future research

For future research, it would be valuable to delve deeper into understanding researchers' proficiency, comprehension, skills, and attitudes necessary for effective community-engaged research. Additionally, there is a need to investigate the specific roles communities can play in each stage of the research process, establishing clear parameters for their involvement. Furthermore, exploring effective strategies to reinforce research-related guidelines and regulations is imperative, as community engagement is not only a privilege but also a community right.

## Supporting information

**S1 Checklist. PRISMA 2020 checklist.**
(DOCX)

**S2 Checklist. JBI critical appraisal checklist of qualitative research.**
(DOCX)

**S3 Checklist. Data extraction checklist or tool for qualitative research.**
(DOCX)

**S1 Table. MEDLINE (Via PubMed) search strategy for community engagement in research (Last search date: July 10, 2023).**
(DOCX)

**S2 Table. Characteristics of included studies.**
(DOCX)

**S3 Table. Summary of findings on community engagement activities/experiences in research.**
(DOCX)

**S4 Table. Summary of findings related to challenges and their mitigation strategies on community engagement in research.**
(DOCX)

**S1 Text. Excluded Studies with the reason for exclusion.**
(DOCX)

## Acknowledgments

Our special thanks go to the invaluable contribution of our project review panel members Nicole Bergen, University of Ottawa, Canada, and Dr. Mirgissa Kaba, Addis Ababa University, Ethiopia.

## Author Contributions

**Conceptualization:** Zewdie Birhanu Koricha, Yosef Gebreyohannes Abraha, Sabit Ababor Ababulgu, Gelila Abraham, Sudhakar Morankar.

**Data curation:** Yosef Gebreyohannes Abraha, Sabit Ababor Ababulgu.

**Formal analysis:** Zewdie Birhanu Koricha, Yosef Gebreyohannes Abraha, Sabit Ababor Ababulgu.

**Funding acquisition:** Zewdie Birhanu Koricha.

**Investigation:** Zewdie Birhanu Koricha, Yosef Gebreyohannes Abraha.

**Methodology:** Zewdie Birhanu Koricha, Yosef Gebreyohannes Abraha, Gelila Abraham.

**Project administration:** Zewdie Birhanu Koricha.

**Software:** Yosef Gebreyohannes Abraha, Sabit Ababor Ababulgu.

**Supervision:** Sudhakar Morankar.

**Validation:** Zewdie Birhanu Koricha, Yosef Gebreyohannes Abraha, Sabit Ababor Ababulgu, Gelila Abraham, Sudhakar Morankar.

**Visualization:** Yosef Gebreyohannes Abraha, Sabit Ababor Ababulgu.

**Writing – original draft:** Zewdie Birhanu Koricha, Yosef Gebreyohannes Abraha, Sabit Ababor Ababulgu, Gelila Abraham, Sudhakar Morankar.

**Writing – review & editing:** Zewdie Birhanu Koricha, Yosef Gebreyohannes Abraha, Sabit Ababor Ababulgu, Gelila Abraham, Sudhakar Morankar.

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
