## [Decision Letter · Decision Letter 0]

16 Nov 2023

PGPH-D-23-01748

Community Engagement in Research Addressing Infectious Diseases of Poverty in Sub-Saharan Africa: A Qualitative Systematic Review

Dear Abraha,

Thank you for submitting your manuscript to PLOS Global Public Health. After careful consideration, we feel that it has merit but does not fully meet PLOS Global Public Health’s publication criteria as it currently stands. Therefore, we invite you to submit a revised version of the manuscript that addresses the points raised during the review process.

We look forward to receiving your revised manuscript.

Kind regards,

Collins Otieno Asweto, PhD

Academic Editor

Journal Requirements:

Reviewers' comments:

Reviewer's Responses to Questions

**Comments to the Author**

1. Does this manuscript meet PLOS Global Public Health’s publication criteria? Is the manuscript technically sound, and do the data support the conclusions? The manuscript must describe methodologically and ethically rigorous research with conclusions that are appropriately drawn based on the data presented.

Reviewer #1: Yes

Reviewer #2: Yes

2. Has the statistical analysis been performed appropriately and rigorously?

Reviewer #1: Yes

Reviewer #2: N/A

3. Have the authors made all data underlying the findings in their manuscript fully available (please refer to the Data Availability Statement at the start of the manuscript PDF file)?

Reviewer #1: Yes

Reviewer #2: Yes

4. Is the manuscript presented in an intelligible fashion and written in standard English?

Reviewer #1: Yes

Reviewer #2: Yes

5. Review Comments to the Author

Reviewer #1: General Assessment:

The manuscript titled "Community Engagement in Research on Infectious Diseases of Poverty in Sub-Saharan Africa: A Qualitative Systematic Review" presents a comprehensive and well-structured study that investigates community engagement experiences, challenges, and mitigation strategies in research related to infectious diseases of poverty in Sub-Saharan Africa (SSA). The authors have followed a clear and transparent methodology, employing established frameworks and tools. Below are the key aspects of the paper's strengths and areas for improvement:

Strengths:

Clear Research Questions and Objectives: The paper effectively establishes its research questions and objectives, which are critical for guiding the study. The questions are relevant to the research context and well-structured.

Comprehensive Methodology: The paper's methodology is robust and systematic. The use of the Joanna Briggs Institute methodology, the PICo framework, and the WHO community engagement framework provides a strong foundation for conducting a qualitative systematic review.

Transparency and Replicability: The paper demonstrates a commitment to transparency and replicability by providing a link to the a priori protocol on the Open Science Framework (OSF). This enhances the trustworthiness of the study.

Appropriate Search Strategy: The three-step search strategy, including both quantitative and qualitative data, is comprehensive and methodologically sound. The decision to restrict the search to studies published since January 2005 is justified to capture current practices.

Conflict Resolution: The authors' approach to resolving disagreements between reviewers during the study selection process is well-documented and aligns with best practices in systematic reviews.

Thematic Synthesis: The shift from a mixed-method systematic review to a thematic synthesis approach is justified based on the available data. The use of Hanacek's guide and the WHO community engagement framework for data synthesis is a valuable addition.

Areas for Improvement:

Clarity of Reporting: While the paper is generally well-written, some sections could benefit from improved clarity. For instance, the explanation of the thematic synthesis approach and its application could be more detailed for readers who may not be familiar with this methodology.

Visual Aids: While there are existing figures and tables, the paper would benefit from additional visual aids that could help summarize key findings or illustrate the application of the WHO community engagement framework.

Consideration of Language: The paper should undergo a thorough proofreading for language and grammar to ensure clarity and professionalism.

Overall, the paper represents a commendable effort to investigate community engagement in research in SSA. It offers valuable insights and recommendations for future research. Addressing the areas for improvement outlined above can enhance the clarity and impact of the paper, making it a more valuable contribution to the field.

Reviewer #2: Dear Authors,

This is an interesting topic and approach to systematic review. You did a thorough and extensive work.

I want to draw your attention to the following points and effect your corrections.

INTRODUCTION

Lines 100 - 104: Review this area for clarity, especially this sentence "LMICs i.e., communities are often approached for individual research projects where only a limited contact is possible, and making research on infectious diseases of poverty has a little contribution to health improvements and capacity building of individuals and institutions [24-33]"

Lines 105-106: Perhaps remove the word "especially" from " ... is extremely weak especially in LMICs"... .

Also, spell out the World Health Organization instead of using WHO at this first use.

Line 111: "achieve intended outcomes" should read "achieving intended outcomes"

METHODS

Line 128: This URL is inactive. Please review it.

Line 144: Please review this statement "... experiences in at least in one of ...

Line 225: Please review this statement it appears clumsy as the word "higher" does not equate to "entire".

"The higher the score implies community was engaged in the entire research process and vice versa."

Line 126: I had anticipated seeing the full meaning of JBI here or somewhere in the methods.

Line 238: Apply consistency in the use of natural numerals. You should use "3" for consistency in this sentence "... databases, three from reference lists and 26 from organizational websites).

RESULTS

Line 283: Republic should have an upper case "R" in Democratic republic of Congo.

Lines 366 - 368: Kindly review the part that says "and proposed to" in this expression. "The problem was then identified i.e., high prevalence of Schistosoma among schoolchildren, and proposed to investigate sustainable ways to prevent the disease [55]."

Lines 389 - 390: Review this sentence and perhaps add an "in" before "community".

"The orphaned and separated children in Kenya were engaged the community throughout the research project."

Line 470: Review and perhaps add "that" after "described". A study by Freudenthal et al (2006) [55] described pupils were involved in a household

Line 485: Please review this line and perhaps add "to" after "understand" in this part of the following part of the sentence. "....were carried out understand the communities."

Lines 692 - 693: Please review this expression and punctuation "The peer navigators discuss the vignettes and co-create the Thetha Nami (‘talk to me’) an intervention that included …". Please insert a coma(,) after the bracket.

Line 730: Please write out AMP in full

Lines 841 - 842: The use of "impartially" in this sentence ("...representativeness and impartiality of advisory board, lengthy ...) does not follow the concept described here or is probably not well explained in section 2.3 below on lines 911 - 920.

Line 898: The phrase "They fill those gaps [69]." did not carry any useful meaning.

Line 911 - 920: See lines 841 - 842 above.

Line 948 - 952: This paragraph is still in the original primary authors' "voice". Perhaps review and follow the general writing pattern of the review.

Lines 959 - 961: This section left me wondering if it is a "recommendation" that came from the reviewed article because you established that the researchers encountered the stated challenge, but this aspect is not clear if they resolved it or only recommended how future researchers should resolve similar situations. "These types of challenges can be e addressed by expanding the interests and activities of academics beyond the campus environment, offering the opportunity for research for the integration of theory and practice [67]."

DISCUSSION

Line 1091: Review the use of the word "translator" in this sentence "collectively translator, identify, filter, interpret, adapt, … ". I suspect that you meant “translate”

Line 1102: Review and insert "by" after "study" in this sentence "In fact, a study Freudenthal et al (2006)"

Line 1130: Please review this sentence "namely informing and consulting leaving communities without empowerment." It could read “namely informing and consulting thereby leaving communities without empowerment."

Line 1135 - 1137: Punctuation would normally come after the in-text citations. There appear to be a mix of CSE and Harvard citation style across the entire document. Kindly unify citation style and address ALL in-text citations the article using a single style.

Line 1156 - 1161: Kindly review this paragraph. Suggestions are in double square brackets [[]].

"balancing community needs and expectations [1156 9,58]. However, cautions [caution] must be made while using the community leaders and CAB [[insert “as”]] they could exert undue influence on the local community in favour of participation in the research under consideration, especially during the consenting and assenting process [66]. Thus, whenever community leaders, CBA [[change to CAB]], or influential community leaders [[change to members]] participate in the research it is important to provide effective training and capacity building to minimize such pressure [12].

Lines 1161 - 1163: The sentence "Limited general literacy in general and research literacy in particular combined with language barriers are persistent challenges; [12]” should perhaps be written as "Limited general literacy and research literacy, combined with language barriers, is a persistent challenge [12].

Line 1173: Please, review and correct this sentence "Thus, we might be missed out some experiences reported quantitatively." to "Thus, we might have missed out some experiences reported quantitatively."

Lines 1186 - 1189: Review this statement "On top of this, almost all the research projects engaged the community at the lower level of engagement (i.e., informing or consulting the community at some point in the research process) suggesting the importance of integrating communities in the entire research cycle."

It could be written as "Most research projects only involved the community at a low level (i.e., informing or consulting them at some point) and showed the need to integrate communities throughout the research cycle."

CONCLUSION

Lines 1190 - 1191: You can delete the “yet” since it is redundant and implies a contrast that is already expressed by “even though” in the expression "Even though a complete picture of community engagement in the research process is mostly lacking, yet useful insight and lessons are drawn from this review."

It could be rewritten as "Even though some details of how communities are engaged in the research process are lacking, useful insights and lessons are drawn from the review."

FUNDING

Lines 1254 - 1256: review this statement and express in past tense. "... will not 

---

## [Decision Letter · Decision Letter 1]

14 Mar 2024

PGPH-D-23-01748R1

Community Engagement in Research Addressing Infectious Diseases of Poverty in Sub-Saharan Africa: A Qualitative Systematic Review

Dear Yoseph Gebreyohannes Abraha,

Thank you for submitting your manuscript to PLOS Global Public Health. After careful consideration, we feel that it has merit but does not fully meet PLOS Global Public Health’s publication criteria as it currently stands. Therefore, we invite you to submit a revised version of the manuscript that addresses the points raised during the review process.

We look forward to receiving your revised manuscript.

Kind regards,

Hemant Deepak Shewade, MBBS MD PhD

Academic Editor

Journal Requirements:

Additional Editor Comments (if provided):

There were three reviewers that reviewed your initial submission: Reviewer 1, Reviewer 2 and Additional reviewer. Response to reviewer 1 and 2 along with the corresponding changes in the manuscript are satisfactory.

I see that the response to "Additional reviewer" is not point by point. Many comments made by the "additional reviewer" are valid but it is not sure if the authors have taken them on board or not.

I request the authors to have a relook at the comments made by this reviewer mentioned as "additional reviewer", consider the comments one by one (revise the manuscript accordingly) and respond 'point by point'.

Kindly resubmit

In addition kindly note the minor comments made by reviewer 3 during this round of review.

Reviewers' comments:

Reviewer's Responses to Questions

**Comments to the Author**

1. If the authors have adequately addressed your comments raised in a previous round of review and you feel that this manuscript is now acceptable for publication, you may indicate that here to bypass the “Comments to the Author” section, enter your conflict of interest statement in the “Confidential to Editor” section, and submit your "Accept" recommendation.

Reviewer #3: All comments have been addressed

2. Does this manuscript meet PLOS Global Public Health’s publication criteria? Is the manuscript technically sound, and do the data support the conclusions? The manuscript must describe methodologically and ethically rigorous research with conclusions that are appropriately drawn based on the data presented.

Reviewer #3: Yes

3. Has the statistical analysis been performed appropriately and rigorously?

Reviewer #3: Yes

4. Have the authors made all data underlying the findings in their manuscript fully available (please refer to the Data Availability Statement at the start of the manuscript PDF file)?

Reviewer #3: Yes

5. Is the manuscript presented in an intelligible fashion and written in standard English?

Reviewer #3: Yes

6. Review Comments to the Author

Reviewer #3: Thank you for the opportunity to review this excellent article. It is exhaustive, well-researched, and actionable.

The article is quite lengthy and I worry that this would limit the authors' ability to share findings in a cogent manner. For the results, section 1 is notably longer than section 2 because the authors have provided extensive detail. It would be more appropriate to offer succinct summaries of the overall themes as they are supported by the research, rather than summarizing each research endeavor, as was done successfully in section 2 of the results.

A minor point is that there are several areas of proofreading that need to be completed to ensure fluency of the language (notably in the discussion).

7. PLOS authors have the option to publish the peer review history of their article (what does this mean?). If published, this will include your full peer review and any attached files.

**Do you want your identity to be public for this peer review?** For information about this choice, including consent withdrawal, please see our Privacy Policy.

Reviewer #3: **Yes: **Ramya Sampath, MD

---

## [Editor Report · Decision Letter 2]

9 Apr 2024

Community Engagement in Research Addressing Infectious Diseases of Poverty in Sub-Saharan Africa: A Qualitative Systematic Review

PGPH-D-23-01748R2

Dear Yoseph Gebreyohannes Abraha,

We are pleased to inform you that your manuscript 'Community Engagement in Research Addressing Infectious Diseases of Poverty in Sub-Saharan Africa: A Qualitative Systematic Review' has been provisionally accepted for publication in PLOS Global Public Health.

Best regards,

Hemant Deepak Shewade, MBBS MD PhD

Academic Editor